# Susceptibility Assessment of Landslides in the Loess Plateau Based on Machine Learning Models: A Case Study of Xining City

Li He [1,2,*], Xiantan Wu [3], Zhengwei He [3], Dongjian Xue [3], Fang Luo [3], Wenqian Bai [3], Guichuan Kang [3], Xin Chen [3] and Yuxiang Zhang [3]

1   Key Laboratory of the Northern Qinghai–Tibet Plateau Geological Processes and Mineral Resources, Xining 810000, China
2   College of Tourism and Urban-Rural Planning, Chengdu University of Technology, Chengdu 610000, China
3   College of Earth Sciences, Chengdu University of Technology, Chengdu 610059, China; wsmv2019@163.com (X.W.); hzw@cdut.edu.cn (Z.H.); dj101@sina.com (D.X.); 2022010019@stu.cdut.edu.cn (F.L.); baiwenqian@stu.cdut.edu.cn (W.B.); kangguichuanan@stu.cdut.edu.cn (G.K.); chenxin_cx@stu.cdut.edu.cn (X.C.); cdutzyx1996@126.com (Y.Z.)
*   Correspondence: heli2020@cdut.edu.cn

**Abstract:** Landslide susceptibility assessment can effectively predict the spatial distribution of potential landslides, which is of great significance in fields such as geological disaster prevention, urban planning, etc. Taking Xining City as an example, based on GF-2 remote sensing image data and combined with field survey data, this study delineated the spatial distribution range of developed landslides. Key factors controlling landslides were then extracted to establish a landslide susceptibility assessment index system. Based on this, the frequency ratio (FR), random forest (RF), support vector machine (SVM), and artificial neural network (ANN) models were applied to spatially predict landslide susceptibility with slope units as the basis. The main results are as follows: (1) The overall spatial distribution of landslide susceptibility classes in Xining City is consistent, but the differences between different landslide susceptibility classes are significant. (2) The high-susceptibility area predicted by the FR-RF model is the largest, accounting for 15.48% of the total study area. The prediction results of the FR-ANN and FR-SVM models are more similar, with high-susceptibility areas accounting for 13.96% and 12.97%, respectively. (3) The accuracy verification results show that all three coupled models have good spatial prediction capabilities in the study area. The order of landslide susceptibility prediction capabilities from high to low is FR-RF model > FR-ANN model > FR-SVM model. This indicates that in the study area, the FR-RF model is more suitable for carrying out landslide susceptibility assessment.

**Keywords:** landslide susceptibility; machine learning; Xining City; loess area

## 1. Introduction

Since the onset of the Quaternary Period, under the influence of tectonic processes, the Loess Plateau has witnessed intermittent episodes of uplift [1]. These geological forces incited profound river incision, sculpting a distinctive loess topography typified by gullies and ravines. Vast and continuous deposits of loess material span regions encompassing Qinghai, Gansu, Ningxia, and Shanxi; the total coverage area of loess is about $4.4 \times 10^5$ km$^2$ [2]. Due to the distinctive properties of loess, including vertical joints and collapsibility, coupled with the intensified precipitation resulting from the continental monsoon climate prevalent in the Loess Plateau, loess landslides occur frequently [3,4]. The Loess Plateau region holds pivotal significance as a major population center in China and a cornerstone for coal fossil energy production. Loess landslides are widespread in the region with complex conditions, which severely impact the life and property security of local residents and pose threats to national

energy security. Despite the substantial attention and effort dedicated to disaster prevention and mitigation by relevant authorities, the region continues to bear the brunt of catastrophic and often fatal landslides. Over the past two decades, advancements in modern Earth observation technologies and the widespread adoption of numerous machine learning models have brought new opportunities for landslide monitoring and early warning [5–7]. Landslide susceptibility assessment can clearly present the spatial probability of landslide hazards. The implementation of landslide susceptibility assessments carries profound implications, not only for the safeguarding of human lives but also for the judicious selection of sites for engineering construction projects.

Landslide susceptibility assessment is an important method for predicting the spatial distribution of landslide hazards. A multitude of scholars have undertaken such assessments through a combination of qualitative and quantitative approaches, anchored in the consideration of influential factors encompassing topography, climate, hydrology, and human activities within the study area. These assessments are based on mathematical probability statistical models or machine learning models to reflect the probability and potential spatial distribution of regional landslides [8–11]. The methodologies for landslide susceptibility assessment can be categorized into two overarching streams: statistical methods and machine learning methods. The statistical methods mainly include certainty factor (CF), information value (IV), weight of evidence (WOE) [12], and frequency ratio (FR) methods [13]. For instance, Xiong [14] conducted an assessment of landslide susceptibility in Puge County, Sichuan Province, employing diverse statistical coupling models, and discerned that the WF-LR model had the highest evaluation accuracy. Fundamentally, landslides are complex nonlinear processes. Compared with traditional statistical models, machine learning models such as random forest (RF) [15], artificial neural network (ANN) [16], support vector machine (SVM) [17], and naive Bayes (NB) [18] are more suitable for dealing with nonlinear problems and can better extract deep data characteristics. Therefore, they are widely applied in landslide susceptibility assessment.

Moreover, several scholars combined statistical models with machine learning models to improve the accuracy of landslide susceptibility assessment. For instance, Ma [19] established a database of 14 influencing factors and used the FR model to process the raw landslide factor data. They combined the RF and FR models to conduct a landslide susceptibility assessment in Lueyang County, finding that landslide hazards were negatively correlated with linear factors. Xia [20] established landslide susceptibility zoning maps of the Three Gorges Reservoir area based on SVM, ANN, and SVM-ANN models, respectively. The study results revealed that the prediction accuracy of individual SVM and ANN models were lower than the coupled SVM-ANN model. The SVM-ANN model is more suitable for practical applications of landslide disaster risk analysis. Research has shown that each model has its advantages and disadvantages, and the predictive ability of the same model varies in different research areas. In areas with complex geological conditions and frequent disasters, the landslide susceptibility results obtained by relying on a single model may have potential uncertainty [21]. Therefore, the combination and comparison of different methods can improve the accuracy of landslide susceptibility prediction in complex areas and provide more scientific guidance and suggestions for landslide disaster prevention and control.

The Loess Plateau has a complex geological environment and frequent human activities, making it a hotspot for landslide susceptibility assessment research. Xining City, an important central city in northwest China, is located in the Loess Plateau. In recent years, geological disasters mainly induced by landslides have frequently occurred in Xining, causing huge losses of lives and property [22,23]. Although some previous studies conducted landslide susceptibility assessments for the entire Loess Plateau region [24,25], there has been relatively little research focused on Xining City. In particular, studies using various coupled models are lacking. Therefore, based on remote sensing images from Gaofen-1 satellite and Google Earth combined with field investigation, this study delineated landslide locations and extents in Xining City. Using slope units as the assessment units, statistical

methods were combined with machine learning methods, connecting the frequency ratio (FR) model with the random forest (RF) model, artificial neural network (ANN) model, and support vector machine (SVM) model, to rapidly and accurately assess landslide susceptibility in Xining City. The accuracy of the calculation results from the three models was validated, and the advantages and disadvantages of different models were analyzed. Landslides are one of the most common types of natural disasters in the Loess Plateau region, and their potential threats have a significant impact on human society and the environment. This study employs machine learning algorithms to assess the susceptibility to landslides, aiming to assist Xining City in gaining a more comprehensive understanding and effective management of landslide risks. This endeavor not only contributes to the protection of lives and property of residents but also supports the maintenance of societal sustainability. Furthermore, the research findings provide crucial information for urban planning and land use decisions. Urban planners can utilize this information to avoid construction in high landslide risk areas, thereby promoting sustainable land use, reducing disaster-induced losses, and enhancing the city's sustainability. The rest of this paper is structured as follows: Section 2 details the specifics of the study area. Section 3 focuses on data sources, methods, and models. In Section 4, we introduce the preparation of landslide susceptibility assessment from four aspects: landslide hazard identification, slope unit extraction, index system construction, and determination of landslide samples. Finally, Section 5 is about the results and analysis of landslide susceptibility assessment.

## 2. Overview of the Study Area

The study area is located in the western part of the Loess Plateau, in the transition zone between the Qinghai–Tibet Plateau and the Loess Plateau (Figure 1). It is situated in the midstream valley basin of the Huangshui River, with an area of approximately 490 km$^2$. The Huangshui River runs from west to east through the entire Xining City area. Under the control of fluvial erosion and deposition as well as tectonic movements, the study area has developed erosion tectonic low mountains and hills along with erosion–deposition river valleys and plains. This makes the topography of the study area show a pattern of high surrounding areas and a low center.

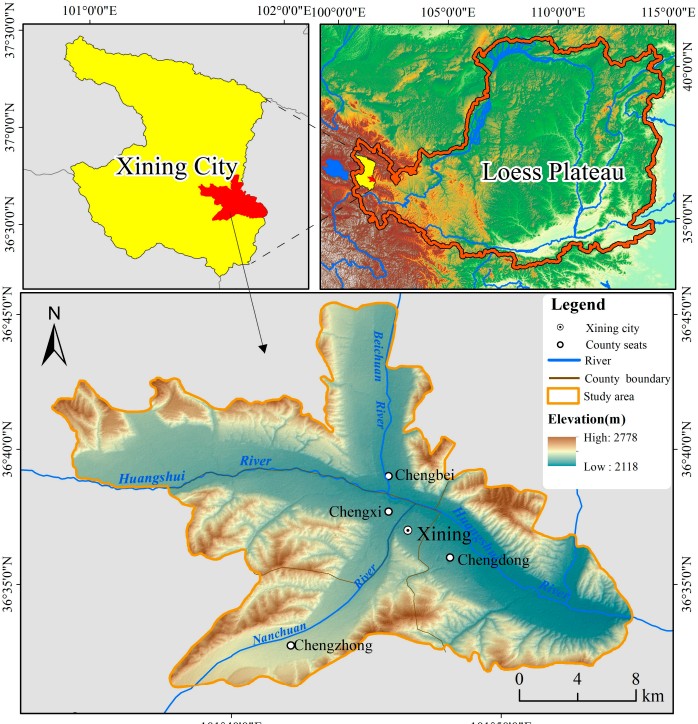

**Figure 1.** Location of study area.

Xining City has a semiarid continental highland climate, characterized by large diurnal temperature variations, long sunshine duration in summer, and mean annual precipitation around 495 mm (Figure 2). As seen in Figure 2, there are significant seasonal differences in precipitation within the study area, making the region susceptible to heavy rainfall events and thus increasing the likelihood of landslide development. Figure 3 illustrates the spatial distribution of annual average precipitation within the study area. The disparity between the highest and lowest precipitation values is 14 mm, indicating a relatively modest overall spatial variation in precipitation. The primary fold structure in the study area is the Huangshui anticlinal axis. The major fault zones along the margins of the basin include the Larishan Fault Zone, Dabashan Fault Zone, and Riyueshan Fault Zone [26], which are all active fault zones and provide the geological background conditions for the development of landslide hazards in the study area. Moreover, the study area is located in the loess region [27], where loess has weak resistance to erosion. Under the influence of factors like river channel scouring, slope excavation, and agricultural irrigation, the stability of sloping soil masses has been compromised and soil strength weakened, which has consequently induced numerous landslides, collapses, and other geological disasters.

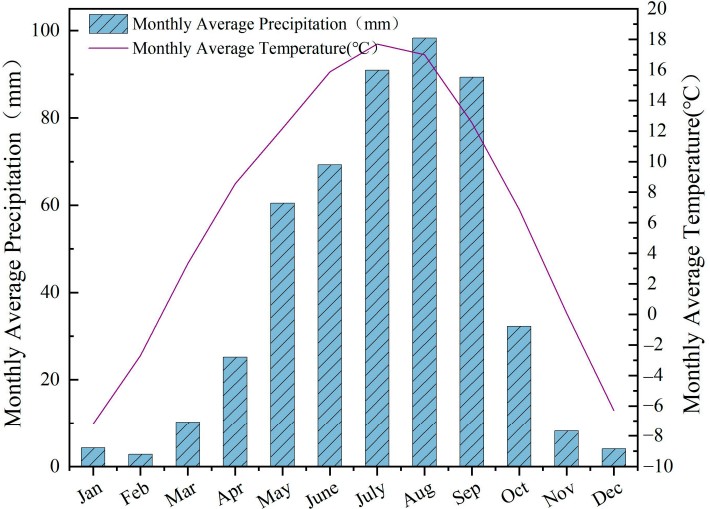

**Figure 2.** Monthly average precipitation (mm) and monthly average temperature (°C) in Xining.

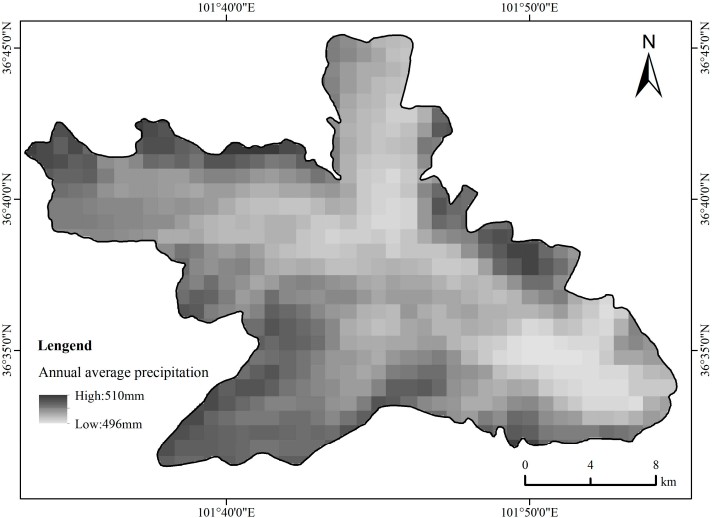

**Figure 3.** Annual average precipitation (mm) in study area.



## 3. Materials and Methods

### 3.1. Data Source and Data Preparation

Two scenes of high-resolution remote sensing imagery covering Xining City were utilized in this study, acquired by the Gaofen-2 satellite on 8 February 2019 and 1 March 2022 (Row 145–149, Path 38–40). The spatial resolution is 1 m panchromatic and 4 m multispectral for these images. Digital elevation model (DEM) data were sourced from the 30 m resolution global DEM acquired by the Advanced Land Observing Satellite "ALOS-1". Based on the DEM, the slope, aspect, and curvature were derived for the study area. Additionally, the relative slope position (RSP) and topographic wetness index (TWI) factors were computed from the DEM using SAGA GIS software (7.6.2, SAGA-GIS, Hamburg, Germany). Based on the basic geological background information provided by the 1:200,000 geological map combined with the GF-2 remote sensing imagery, the faults and lithologies in the study area were deciphered using remote sensing interpretation and field investigation methods [28,29]. Road data were sourced from the OpenStreetMap platform, and the distances from roads, water systems, and faults were generated based on the ArcGIS software (10.8.2, Esri, California, United States) platform. Precipitation data were obtained from the China Meteorological Data Network (http://data.cma.cn) (accessed on 2 December 2022).

### 3.2. Evaluation Method for Landslide Susceptibility

#### 3.2.1. Frequency Ratio

The frequency ratio (FR) is a statistical method that is simple to implement with accurate results; it is widely used in landslide susceptibility mapping and highly compatible with GIS technology [30–34]. An FR of 1 is the average value from the ratio of the area where landslides occurred to the total area. If the probability value is greater than 1, there is a greater susceptibility for landslides and vice versa [30]. The FR model determines the probability of landslide occurrence within different classification intervals for each landslide susceptibility assessment factor, thereby quantifying the influence of each factor on landslides [19]. The formula for calculating the FR is as follows:

$$FR = \sum_{i=1}^{n} \left( \frac{N_i/N}{S_i/S} \right) \tag{1}$$

where:

$FR$–frequency ratio of a certain unit in the research area (/);
$n$–the number of categorized intervals for evaluating factors (/);
$N_i$–number of mapping units containing landslide inventory points in the $i$th class of the factor (/);
$N$–the total number of units with distributed disaster points (/);
$S_i$–the area of the $i$th class for the factor (km$^2$);
$S$–the total area of the study area (km$^2$).

#### 3.2.2. Random Forest

Random forest (RF), originally proposed by Breiman [35], is an ensemble learning method that constructs multiple decision trees for classification. The RF algorithm employs bootstrap sampling to randomly select k training subsets, with replacements from the original training set equal in size to the original sample. Decision trees are fitted on each of the k samples to obtain k classification results. Final predictions are obtained by majority vote of the k decision tree outputs for each record, giving the optimal classification. The random forest model demonstrates robustness to noise and outliers, especially when dealing with very large sample sizes [36,37]. For landslide identification and susceptibility assessment, the RF algorithm has shown strong performance and has been widely utilized by researchers in related studies [38,39].

### 3.2.3. Artificial Neural Networks

Artificial neural networks (ANN) are inference models established on the basis of mimicking the functions and neural system of the human brain [40]. Their structure consists of three parts: an input layer, a hidden layer, and an output layer. They possess powerful capabilities for nonlinear processing. This paper utilizes a multilayer perceptron neural network to conduct landslide susceptibility assessment. The basic idea is as follows: Landslide influencing factors are used as input neurons. Different neurons are interconnected via weights that propagate to the output layer. The weight calculation is expressed as Formula (2) [41]:

$$y_i = f\left(\sum\nolimits_i W_{ij} + b_j\right) \tag{2}$$

where:

$W_{ij}$–the weight connecting neuron *i* and neuron *j* (/);
$b_j$–the bias term (/);
$f$–the activation function (/).

### 3.2.4. Support Vector Machine

The support vector machine (SVM) is a nonlinear classification and regression machine learning method proposed by Cortes and Vapnik in 1995 [42]. The principle is structural risk minimization with a theoretical basis in statistical learning theory. The SVM maps the input variables from the original space to a higher dimensional feature space through various kernel functions in order to find the optimal separating hyperplane. The SVM model has strong generalization abilities and fast convergence rates when dealing with high-dimensional data of small samples [20,24] and has been widely applied in landslide susceptibility assessment. This model is mainly influenced by the kernel width ($\gamma$) and regularization (C) parameters. The grid search method is used to obtain the optimal parameters $\gamma$ and C, which are 2 and 0.4, respectively.

In the Python environment, the frequency ratio (FR) values of each evaluation factor in the sample dataset were input into the RF, ANN, and SVM machine learning algorithms with optimal parameters to make predictions and construct FR-RF, FR-ANN, and FR-SVM models to calculate landslide susceptibility indexes.

### 3.3. Method for Verifying the Accuracy of Evaluation Models

In order to compare the above methods, this paper selects the confusion matrix, accuracy, kappa coefficient, and area under the receiver operating characteristic curve (AUC) to evaluate model performance [43–45].

### 3.3.1. Confusion Matrix

The confusion matrix is a matrix that measures the prediction performance of classification models, presenting algorithm performance in the most fundamental and intuitive way [46]. It contains the number of samples that are correctly or incorrectly classified for each category (Table 1). TP (true positive) means correctly predicting positive samples as positive, FN means false negative, indicating incorrectly predicting positive samples as negative, FP (false positive) means incorrectly predicting negative samples as positive, and TN (true negative) means correctly predicting negative samples as negative.

**Table 1.** Confusion matrix.

|  | **Predicted Landslide** | **Predicted Nonlandslide** | **Sum** |
|---|---|---|---|
| Actual landslide | TP | FN | TP + FN |
| Actual nonlandslide | FP | TN | FP + TN |
| Sum | TP + FP | FN + TN | TP + FN + FP + TN |

### 3.3.2. Kappa Coefficient

The kappa coefficient aims to test whether the model prediction results are consistent with the actual classification results for classification problems. The calculation depends on the confusion matrix, and the calculation formula is as follows [47]:

$$k = \frac{p_o - p_e}{1 - p_e} \tag{3}$$

where:

$P_o$–the sum of the number of correctly classified samples in each class divided by the total number of samples (/).

$P_o$ is also the overall classification accuracy. Assuming the true number of samples in each class is $a_1, a_2 \ldots a_m$ and the predicted number of samples in each class is $b_1, b_2 \ldots b_m$, with a total of n samples. The formula for $P_e$ is

$$p_e = \frac{a_1 b_1 + a_2 b_2 + \cdots a_m b_m}{n^2} \tag{4}$$

### 3.3.3. Receiver Operating Characteristic (ROC)

The receiver operating characteristic (ROC) curve is a method to quantitatively evaluate the prediction accuracy of binary classification models [48]. The method uses a series of true positive rates (TPRs) and false positive rates (FPRs) to plot the curves and calculates the area outside the curve (AUC) to assess the predictive accuracy of the model. When the classification threshold is changed, the classification result of the model changes, which affects the TPR and FPR of different models for different landslide classes. When the classification threshold is lowered, it means that the model correctly identifies landslide units more frequently, which will result in an increase in the TPR and FPR. On the contrary, when the threshold is increased, it leads to a decrease in TPR and FPR. Therefore, TPR and FPR are calculated by constantly changing the classification thresholds to plot the ROC curves, from which the AUC results are calculated.

## 4. Construction of Landslide Susceptibility Evaluation System

### 4.1. Optical Remote Sensing Interpretation of Landslides

Based on Gaofen-2 satellite image data and historical imagery data from Google Earth, landslides in Xining City were visually interpreted. Combining the planar morphology, topographic conditions, and tone and texture characteristics of landslides in the optical imagery, the boundaries of landslides in the study area were preliminarily delineated. We synthesized field investigation data and deleted wrongly interpreted points (human engineering activities, surface erosion); a total of 242 landslides were delineated, with a total area of 2.97 km². The scale of these landslides is mainly small- and medium-sized, and the spatial distribution of landslides is shown in Figure 4.

### 4.2. Extracting Slope Elements Based on the r.slopeunits Method

The evaluation unit is the basis for regional landslide stability assessment, with similar geological and geomorphological characteristics within the same evaluation unit. The slope unit is the basic geomorphological unit for landslide occurrence. This study takes the slope unit as the basic mapping unit for regional landslide susceptibility assessment and extracts slope units using the r.slopeunits method [49,50]. This method is based on the GRASS GIS platform and uses iterative and adaptive algorithms. By inputting DEM, plain data, flow accumulation area threshold (*t*), minimum slope unit area, and other parameters, slope units were obtained. A total of 2072 slope units were calculated in the study area, and the distribution of slope units is shown in Figure 5.

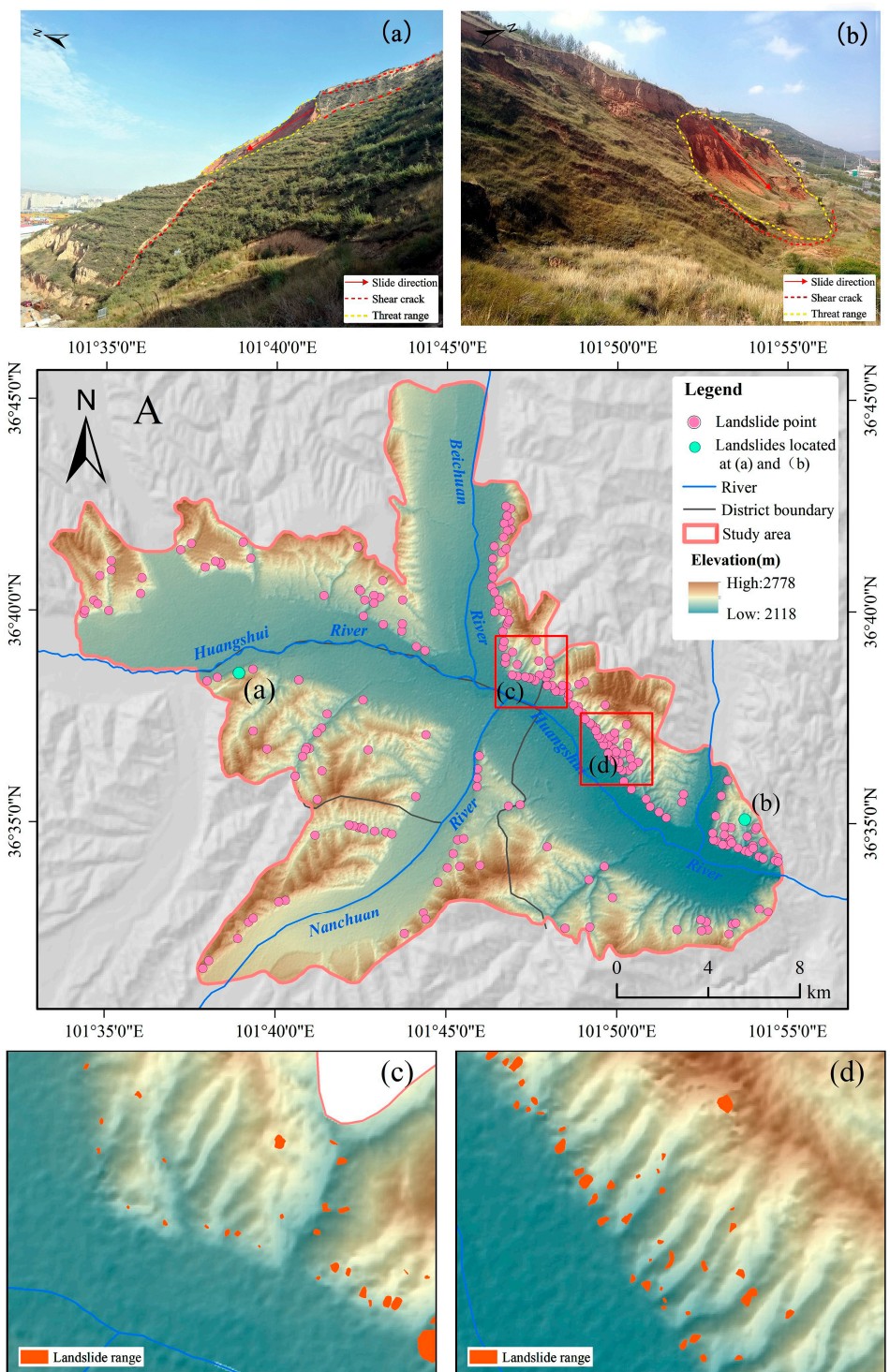

**Figure 4.** Distribution map of landslide sites in the study area: (**a**) landslide located in Zhangjiawan village; (**b**) landslide located in Hanzhuang village; (**c**,**d**) the range of landslide development.

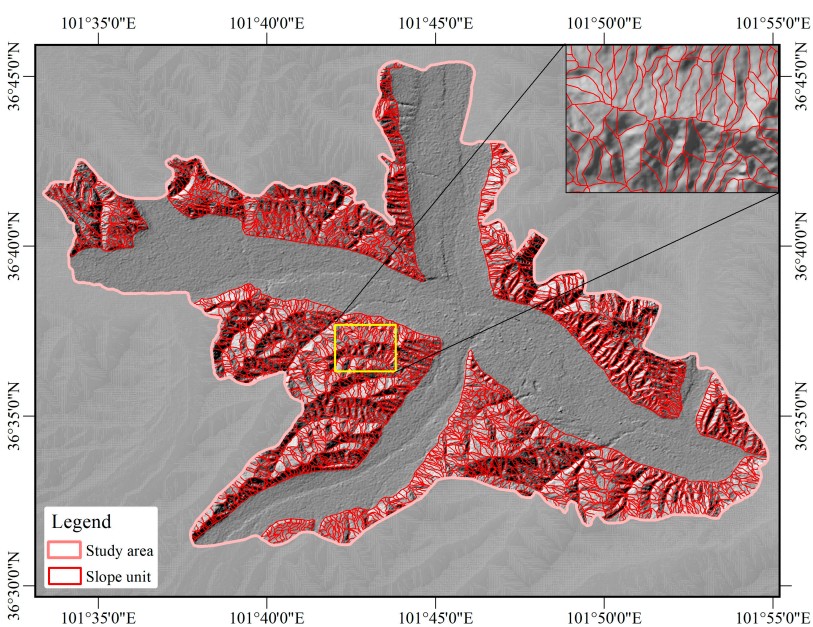

**Figure 5.** Division of slope units in the study area.

### 4.3. Establishment of Susceptibility Evaluation Index

Landslides are the product of the combined effects of multiple factors including geology, geomorphology, climate, etc. Selecting suitable factors is the key to establishing a landslide susceptibility assessment. Based on previous research [8,10,20,51] and the actual conditions of the study area, nine factors were selected to construct the landslide susceptibility assessment index system, including slope, aspect, curvature, topographic wetness index (TWI), relative slope position (RSP), lithology, distance to faults, distance to rivers, and distance to roads. Among them, aspect, curvature, and lithology are discrete types, and the other factors are continuous (Figure 6). It is noteworthy that for the loess area, rainfall is an important factor in triggering landslides. It is undeniable that precipitation is a significant triggering factor for landslide development. The seasonal distribution of precipitation in Xining City is highly uneven, making the region susceptible to heavy rainfall, which in turn reduces slope stability and increases the likelihood of landslide occurrences. Considering the conditions required for precipitation, most precipitation events typically have a wide spatial extent. However, due to the limited size of our study area, covering only 490 km², and the relatively minor spatial variability in precipitation, it has therefore not been included as an evaluation criterion in our study.

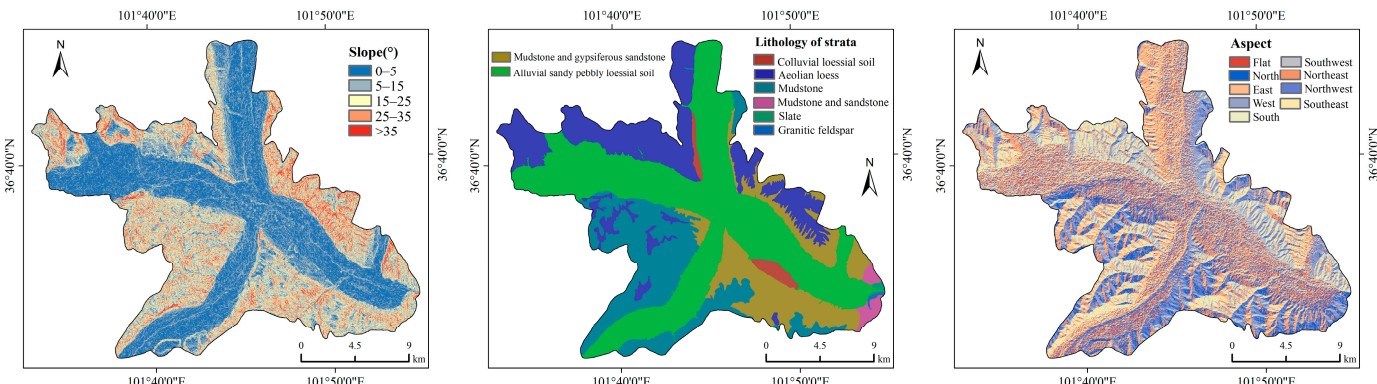

**Figure 6.** *Cont.*

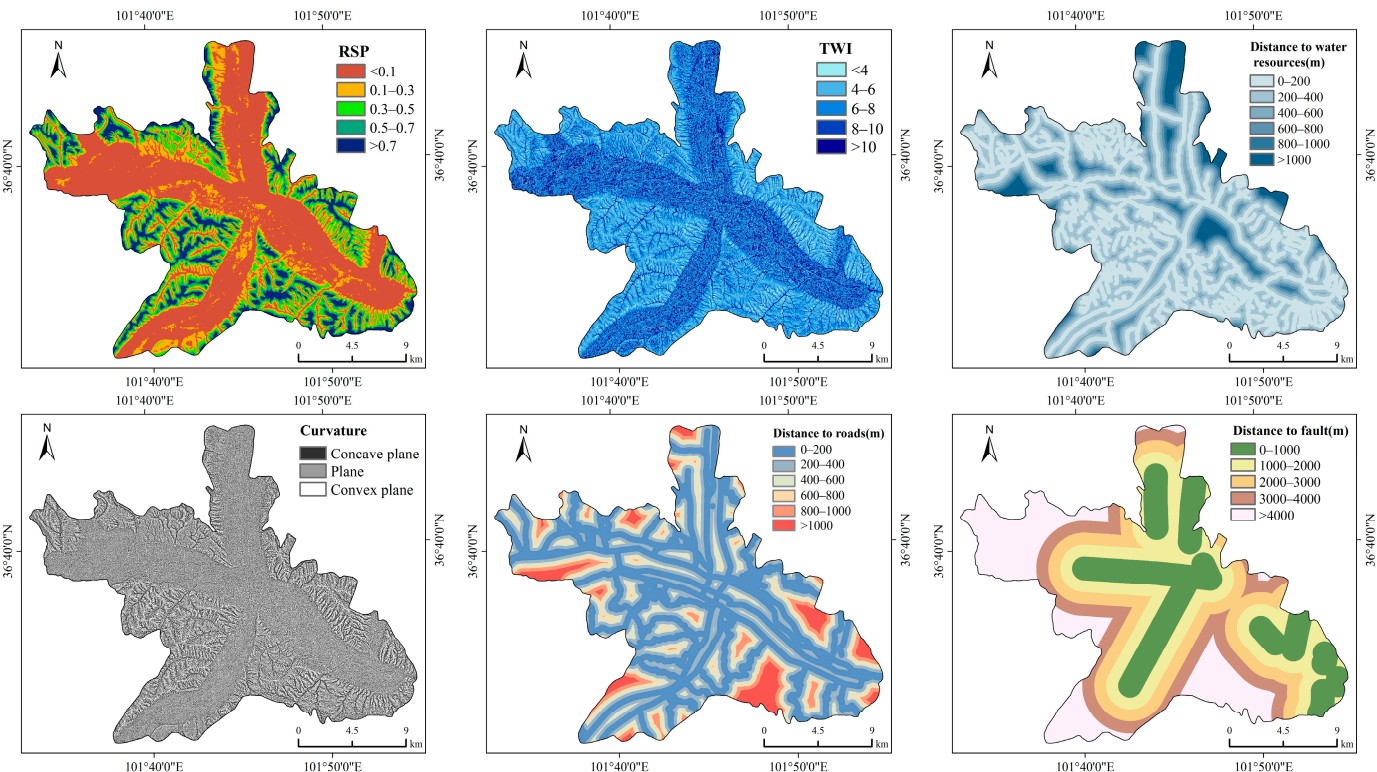

**Figure 6.** Index factors of landslide susceptibility evaluation: RSP, relative slope position; TWI, topographic wetness index.

### 4.4. Identification of Landslide and Nonlandslide Samples

Landslide susceptibility was assessed in Xining City based on the FR-RF, FR-ANN, and FR-SVW models. This study examined cases where the landslide area as a percentage of the slope unit area was greater than 1%, 2%, and 3%, respectively. Since the landslides in the study area were dominated by small- and medium-sized landslides, by comparing with the actual situation in the study area, it was found that the 275 slope units with landslides obtained are the best landslide units when the proportion of landslide area to the area of the slope unit is 1%. The establishment of the location of landslide and nonlandslide samples based on slope units affects the calculation of the frequency ratio model. By comparing the magnitude of the frequency ratios, it is possible to determine which features are strongly correlated with landslide occurrence, which in turn affects the calculation of the landslide susceptibility index. The nonlandslide samples were randomly selected within the very-low- and low-landslide-susceptibility zones constructed by the frequency ratio model. In the landslide and nonlandslide sample datasets, 70% of the data were used to construct the RF, ANN, and SVW models, and 30% were used for model testing. The specific distribution of the sample sets is shown in Figure 7.

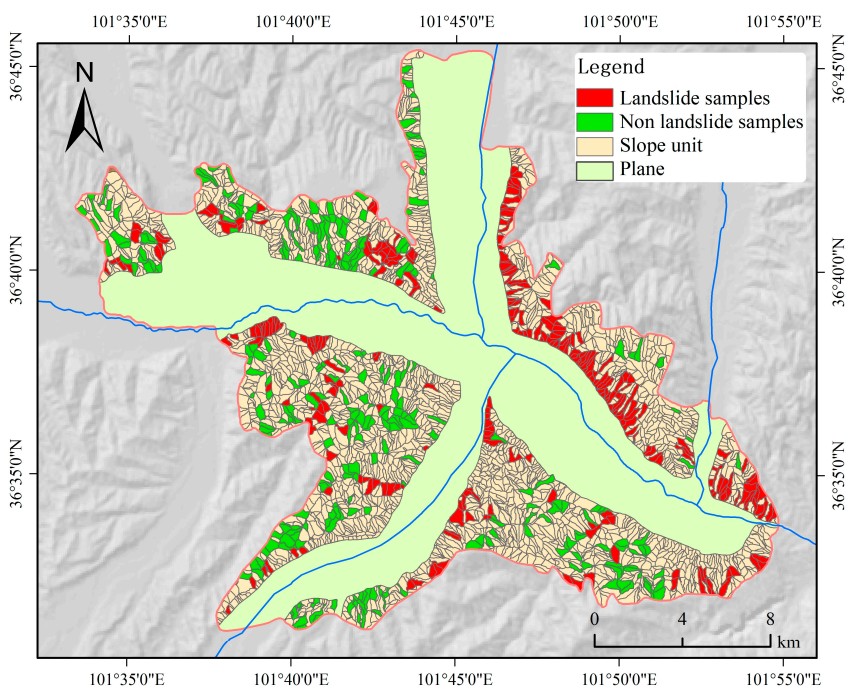

**Figure 7.** Spatial distribution of landslide and nonlandslide samples in the study area.

## 5. Result

### 5.1. Prediction Results and Analysis of Landslide Susceptibility Class in Xining City

Based on FR-RF, FR-ANN, and FR-SVM models, landslide susceptibility indices were calculated and obtained, respectively, and the results are shown in Figure 8. In addition, when extracting slope units, areas with a slope less than 5° were delineated as flat ground, which were identified as very-low-susceptibility areas.

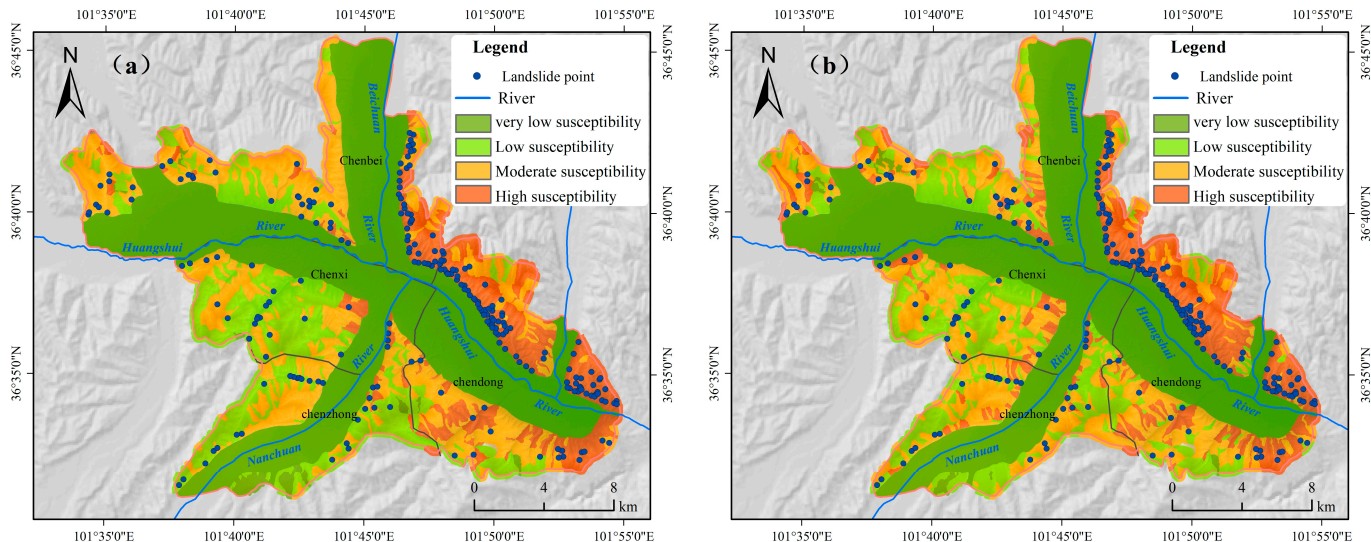

**Figure 8.** *Cont.*

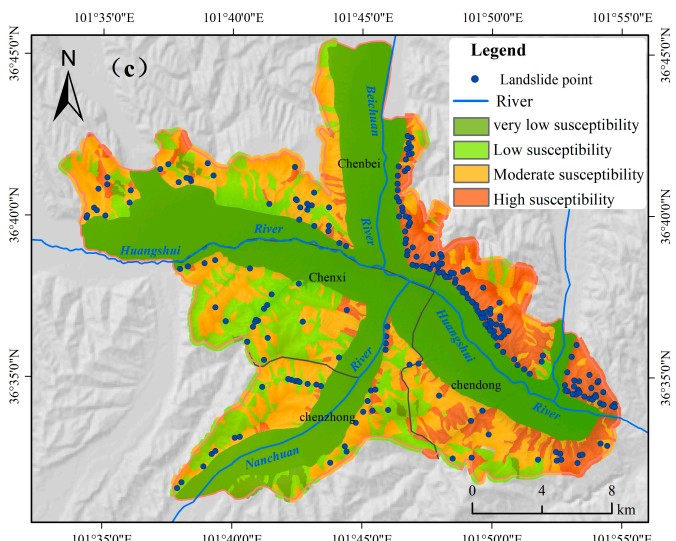

**Figure 8.** Distribution of landslide susceptibility classes based on three models: (**a**) FR-SVM; (**b**) FR-RF; (**c**) FR-ANN.

The overall spatial distribution of landslide susceptibility classes in Xining City is basically consistent. The high-susceptibility areas obtained from all three models are mostly distributed in the northern part of Chengdong District and eastern part of Chengbei District, especially in the transitional zones between the Huangshui Valley and the mountains. The lithology in these areas is dominated by mudstone, sandstone, and gypsum, with interbedded structures. In areas with abrupt topographic changes and dramatic slope increases, the influence of rainfall and groundwater can form weak structural surfaces and induce landslides. Moderate-susceptibility areas are mainly distributed surrounding the high-susceptibility areas, mostly concentrated in Chengbei, Chengxi, and Chengzhong districts. Low-susceptibility areas are mainly concentrated on both sides of the Huangshui River valley.

In order to verify the results of landslide susceptibility distribution obtained based on the three models, we counted the number of landslide sites within the range of the four landslide susceptibility classes and calculated the percentage of the number of landslides within each class, and the results are shown in Table 2. It can be seen that 56.61%, 66.12%, and 60.74% of the landslides are distributed in the high-susceptibility zone, and 30.58%, 21.49%, and 27.69% of the landslides are distributed in the medium-susceptibility zone based on the FR-RF, FR-ANN, and FR-SVM models, respectively. It can be seen that more than 87% of the landslides are distributed in the medium- and high-susceptibility zones, which indicates that the prediction results of landslide susceptibility classes based on the three machine learning models are basically consistent with the actual situation, which also provides a reliable database for landslide disaster risk prevention.

**Table 2.** Percentage of the number of landslide points in the range of four susceptibility classes.

| Susceptibility Classes | FR-SVM | | FR-RF | | FR-ANN | |
|---|---|---|---|---|---|---|
| | Number | Percentage/% | Number | Percentage/% | Number | Percentage/% |
| Very low susceptibility | 0 | 0 | 0 | 0 | 0 | 0 |
| Low susceptibility | 31 | 12.81 | 30 | 12.40 | 28 | 11.57 |
| Moderate susceptibility | 74 | 30.58 | 52 | 21.49 | 67 | 27.69 |
| High susceptibility | 137 | 56.61 | 160 | 66.12 | 147 | 60.74 |

*5.2. Analysis of Differences in Landslide Susceptibility Classes*

The landslide susceptibility class results were obtained based on FR-RF, FR-SVM, and FR-ANN models, and then we counted the area and percentage of different landslide

susceptibility classes, as shown in Table 3.The high-susceptibility area predicted by the FR-RF model has an area of 65.80 km², accounting for 15.48% of the total study area, with a landslide area of 2.18 km² accounting for 76.64% of the total landslide area and a landslide density of 3.32 km²/100 km². The moderate-susceptibility area has an area of 99.18 km², accounting for 23.33% of the total study area, with a landslide area of 0.56 km² accounting for 19.66% of the total landslide area and a landslide density of 0.56 km²/100 km².

**Table 3.** Analysis of the results of different evaluation models' susceptibility partitioning.

| Evaluation Model | Landslide Susceptibility Grades | Area (km²) | Areal Percentage (%) | Landslide Area (km²) | Proportion of Landslide Area (%) | Landslide Density (/100 km²) |
|---|---|---|---|---|---|---|
| FR-RF | Very low susceptibility | 187.63 | 44.13 | 0 | 0 | 0 |
| | Low susceptibility | 72.52 | 17.06 | 0.11 | 3.70 | 0.15 |
| | Moderate susceptibility | 99.18 | 23.33 | 0.56 | 19.66 | 0.56 |
| | High susceptibility | 65.80 | 15.48 | 2.18 | 76.64 | 3.32 |
| FR-ANN | Very low susceptibility | 190.92 | 44.90 | 0 | 0 | 0 |
| | Low susceptibility | 70.81 | 16.65 | 0.15 | 5.21 | 0.21 |
| | Moderate susceptibility | 104.12 | 24.49 | 0.85 | 29.96 | 0.82 |
| | High susceptibility | 59.37 | 13.96 | 1.85 | 64.83 | 3.11 |
| FR-SVM | Very low susceptibility | 189.74 | 44.62 | 0 | 0 | 0 |
| | Low susceptibility | 75.06 | 17.65 | 0.18 | 6.36 | 0.24 |
| | Moderate susceptibility | 105.28 | 24.76 | 0.91 | 32.09 | 0.87 |
| | High susceptibility | 55.14 | 12.97 | 1.75 | 61.55 | 3.18 |

Compared with the FR-RF model, the high-susceptibility area predicted by the FR-ANN model has an area of 59.37 km², accounting for 13.96% of the total study area, with a landslide area of 1.85 km² accounting for 64.83% of the total landslide area and a landslide density of 3.11 km²/100 km². The moderate-susceptibility area has an area of 104.12 km², accounting for 24.49% of the total study area, with a landslide area of 0.85 km² accounting for 29.96% of the total landslide area and a landslide density of 0.82 km²/100 km².

The FR-SVM model has the smallest high-susceptibility area of 55.14 km², with a landslide area of 1.75 km² accounting for 61.55% of the total landslide area and a landslide density of 3.18 km²/100 km². The moderate-susceptibility area has an area of 105.28 km², accounting for 24.76% of the total study area, with a landslide area of 0.91 km² accounting for 32.09% of the total landslide area and a landslide density of 0.87 km²/100 km².

In addition, since the valley plains in the study area are vast with a small probability of landslide occurrence, the very-low- and low-susceptibility areas account for the highest proportion of around 60% in the three coupled models.

## 6. Discussion

### 6.1. Verification of Prediction Accuracy of Different Models

As shown in Figure 9, when using the three coupled models to predict landslide susceptibility, the obtained AUC values are all greater than 0.8, indicating that the models in this study have good spatial prediction capabilities and accuracy. Among them, the FR-RF model has the maximum AUC value of 0.863, followed by the FR-ANN model with an AUC of 0.839, while the FR-SVM model has the minimum AUC value of 0.825.

Figure 10 shows the confusion matrix for each model in the test set, where TPR, FNR, FPR, and TNR refer to true positive rate, false negative rate, false positive rate, and true negative rate, respectively. As shown in Figure 10a, the confusion matrix for the FR-RF model has a TPR of 84%, FNR of 16%, FPR of 20%, and TNR of 80%. The calculated accuracy and kappa coefficient are 0.825 and 0.649, respectively. The confusion matrix for the FR-ANN model has a TPR of 81%, FNR of 19%, FPR of 23%, and TNR of 77%. The calculated accuracy and kappa coefficient are 0.793 and 0.586, respectively (Figure 10b). As shown in Figure 10c, the confusion matrix for the FR-SVM model has a TPR of 79%, FNR of 21%, FPR of 25%, and TNR of 75%. The calculated accuracy and kappa coefficient are 0.773 and 0.545, respectively. The above analysis shows that among the three coupled models, the FR-RF model has the highest kappa coefficient, accuracy, and AUC, indicating it has

the best fitting performance in this study and superior prediction capability compared to the other models.

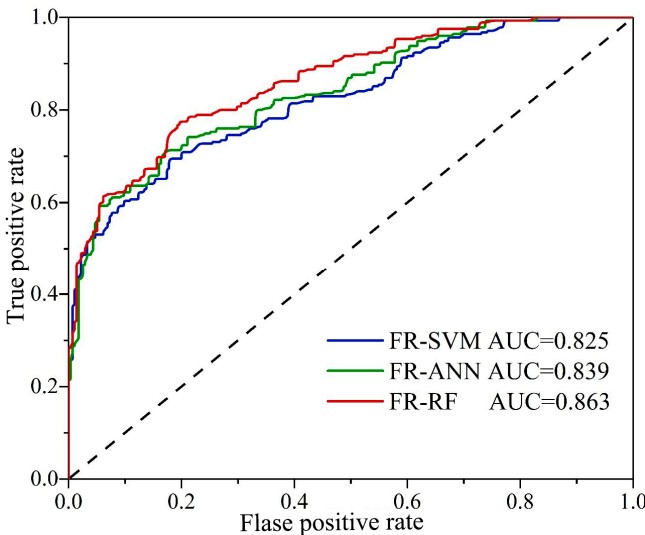

**Figure 9.** Receiver operating characteristic curve.

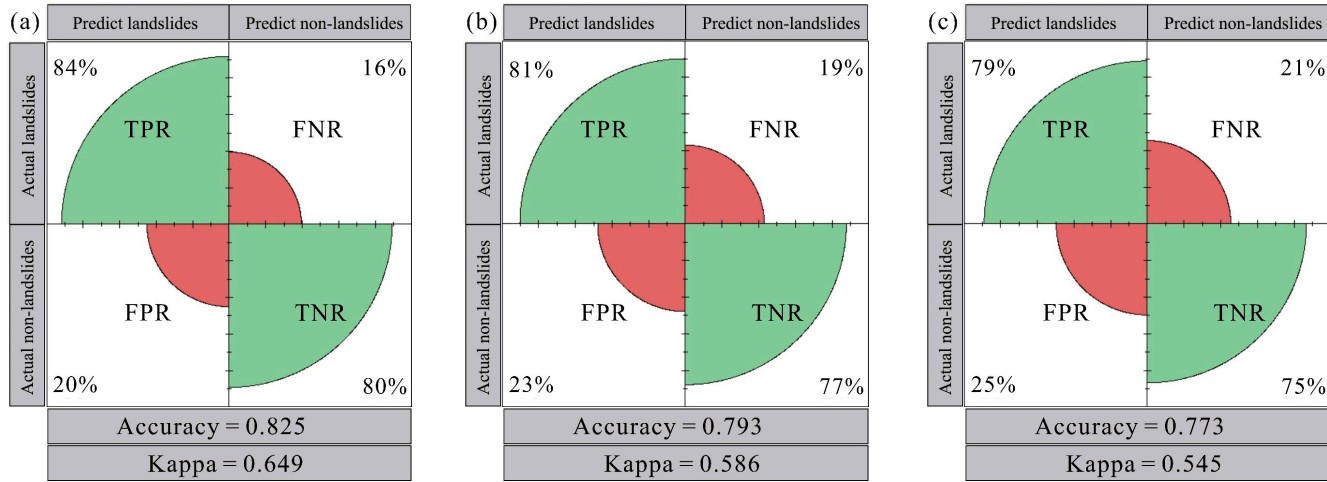

**Figure 10.** Confusion matrix, kappa coefficient, and accuracy values of different evaluation models: (**a**) FR-RF model; (**b**) FR-ANN model; (**c**) FR-SVM model.

## 6.2. Research on Landslide Susceptibility Evaluation Units and Evaluation Models

In the context of selecting assessment units, we introduced slope units as the assessment units. Assessment units play a pivotal role as carriers of landslide disaster-related information, and their selection fundamentally shapes the spatial scope of the assessment. Variations in the chosen assessment units can exert pronounced influences on both the computational efficiency and the accuracy of assessment outcomes. In existing studies on landslide susceptibility assessment, some scholars conducted research using the scales of rectangular grids, irregular grids, and administrative units [52,53].With the deepening of research, the methods for determining assessment units were continuously improved, including grid units, geomorphic units, slope units, and watershed units [54]. Through a literature review, it was found that current studies on landslide susceptibility assessment are mainly based on grid units and slope units [55,56].

The evaluation of landslide susceptibility based on grid units is more convenient in terms of data acquisition and processing. However, it is essential to acknowledge that the resolution of grid units directly impacts the quantity of grid cells, consequently influencing

the determination of an appropriate landslide sample size across distinct intervals of evaluation factors. This approach, based on grid units, may lead to a partial disconnection between landslide occurrences and their underlying geological and geomorphic conditions. Slope is the basic condition for the development of landslides, and the geological lithology, topography, and other conditions within the same slope unit are basically the same [53]. Hence, we chose to employ the r.slopeunits method, embracing slope units as the primary evaluation units for our landslide susceptibility research, which is closely combined with the actual terrain and geology, making the evaluation results closer to reality.

With the development of machine learning models, the problem of data overfitting has been gradually overcome, and they have shown outstanding performance in dealing with nonlinear relationships and generating optimal features. As a result, machine learning models are widely applied in landslide susceptibility assessment research [5,37,43]. This study coupled the FR model with three models—RF, ANN, and SVM—and conducted relevant research based on the FR-RF, FR-ANN, and FR-SVM models. Comparing the evaluation results and accuracy verification of the three models, it was found that although the spatial distributions of landslide susceptibility classes obtained by the three models were quite different, the general distribution patterns were consistent. The FR-RF model yielded the largest high-susceptibility area, accounting for about 15.48% of the study area. The high-susceptibility areas obtained by FR-ANN and FR-SVM were more similar, accounting for 13.96% and 12.97%, respectively. In terms of model accuracy, the area under the curve (AUC) values followed the descending order of FR-RF > FR-ANN > FR-SVM. In summary, while all three models can adequately characterize the regional landslide susceptibility, the FR-RF model demonstrated superior performance in computational efficiency and assessment result accuracy, making it more suitable for landslide susceptibility assessment research in regions with similar geographical environments.

In this study, we used machine learning models, including FR-RF, FR-ANN, and FR-SVM, for landslide susceptibility assessment based on data from Gaofen-1 remote sensing imagery and Google Earth platforms, as well as a comprehensive analysis of field surveys in the loess area of Xining City. By comparing the performance of different models, we arrived at the conclusion that the FR-RF method has more potential in landslide risk assessment, which can help decision makers better understand landslide risk and take appropriate measures. Furthermore, our study serves as a stepping stone for future investigations in loess regions, potentially igniting further exploration and innovation in the field of landslide susceptibility assessment. However, it is undeniable that our study still has certain deficiencies. Given the relatively small size of our study area, we did not incorporate precipitation as a contributing factor in our assessment. Moving forward, it is imperative to place heightened emphasis on the influence of precipitation and groundwater levels on landslide occurrences in our future research endeavors. In addition, we will combine more remote sensing data, such as InSAR data, to further carry out the study of landslide deformation monitoring and landslide susceptibility assessment in the Loess Plateau region so as to provide a more comprehensive scientific basis for the prevention and control of landslide disasters in the Loess Plateau region.

## 7. Conclusions

In the Loess Plateau, landslide disasters occur frequently. In this paper, taking Xining City as an example, based on GF-1 high-resolution remote sensing images and the Google Earth platform combined with a field survey, a total of 242 landslide disasters were delineated in the study area. Slope units were then extracted as evaluation units based on the r.slopeunits method. Landslide susceptibility assessment of the study area was carried out based on FR-RF, FR-ANN, and FR-SVM models, and finally, the prediction results and prediction accuracy of different machine learning models were compared and analyzed. Our experimental results show that the three machine learning models exhibit good performance in landslide susceptibility evaluation and provide strong support for landslide risk management in the study area. We believe that this study is of significant and

important value not only for Xining City but also for other areas with similar geographic environments. The conclusions are as follows:

(1) The overall spatial distribution of landslide susceptibility levels in Xining City is consistent, but there are significant differences between different landslide susceptibility categories. The high-susceptibility areas are mainly concentrated in the plain–mountain transitional areas in the eastern and northern urban districts of Xining City. The medium-susceptibility areas are distributed surrounding the high-susceptibility areas, while the low-susceptibility areas are relatively small. The very-low-susceptibility areas are the largest, with the central plain areas of the region all classified as very-low-susceptibility areas.

(2) The high-susceptibility area predicted by the FR-RF model is the largest, reaching 65.80 km$^2$ and accounting for 15.48% of the total study area. Compared with the FR-RF model, the prediction results of the FR-ANN and FR-SVM models are more similar, with predicted areas of 59.37 km$^2$ and 55.14 km$^2$, accounting for 13.96% and 12.97% of the total area, respectively.

(3) The AUC of the FR-RF model, FR-ANN model, and FR-SVM model in predicting landslide susceptibility is greater than 0.8, indicating that the three coupled models have excellent spatial prediction ability and accuracy. The accuracy verification results show that the prediction ability of landslide susceptibility is in the following order: FR-RF model > FR-ANN model > FR-SVM model. This indicates that in loess areas with similar geographical environments, the FR-RF model is more suitable for landslide susceptibility assessment based on slope units.

**Author Contributions:** L.H.: methodology and writing. X.W.: software and original draft preparation. Z.H. and D.X.: writing—review and editing. F.L., X.C., W.B., G.K. and Y.Z.: field surveys. All authors have read and agreed to the published version of the manuscript.

**Funding:** Supported by Key Laboratory of the Northern Qinghai–Tibet Plateau Geological Processes and Mineral Resources (2019-KZ-02).

**Institutional Review Board Statement:** Not applicable.

**Informed Consent Statement:** Not applicable.

**Data Availability Statement:** The data will be available for review from the corresponding author upon request.

**Acknowledgments:** Thank you to Key Laboratory of the Northern Qinghai–Tibet Plateau Geological Processes and Mineral Resources for providing Gaofen-2 remote sensing image data and funding support for this article. The authors are thankful to all the associated personnel who contributed to this study by any means.

**Conflicts of Interest:** The authors declare no conflict of interest.

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
