# Peer review of "Susceptibility Assessment of Landslides in the Loess Plateau Based on Machine Learning Models: A Case Study of Xining City"

_sustainability, doi:10.3390/su152014761_

Round 1

Reviewer 1 Report

The material and method must be properly and clearly written, complete and clearly presented without ambiguity. The results should present the data obtained from the research without describing the work methods in the same chapter. 

Minor editing of English language required.

Author Response

Dear Reviewer,

Thank you very much for taking the time to review this manuscript. Thank you for your valuable comments and suggestions on this manuscript. We have revised and explained each of your comments and partially optimized the English language of this manuscript, please refer to the detailed responses below and the corresponding revisions highlighted in the resubmission.

Reviewer 2 Report

This is a useful paper on using AI to predict landslide risk. The overall article organization and conclusion are well written, and I would like to ask you to revise.

1. excluded the precipitation factor due to the narrowness of the location, but the added precipitation value needs to be compared (to be considered in future papers)

2. consider future analysis of landslide-prone areas and comparison with sliding finite element analysis at one point.

 -Sliding is often related to groundwater levels, so consider conditions after precipitation.

3. In the conclusion section, only the experimental results have been written. It is necessary to write a conclusion including the author's opinion through this paper.(Required)

Author Response

(The authors gave the same response as above.)

Reviewer 3 Report

The paper introduces a case study focusing on landslide susceptibility assessment in Xining City. The research combines remote sensing data and field surveys to identify existing landslides and extract factors that influence them. Three models (FR-RF, FR-SVM, FR-ANN) are employed to predict susceptibility based on slope units. The FR-RF model achieved the highest prediction performance.

At the end of the Introduction section, briefly summarize the contribution of the paper and outline the paper’s structure for the subsequent sections. For instance, Section 2 describes the study area, Section 3 introduces the data sources and methodology, etc.

Explain more on the ROC curve on how to get the series of TPR and FPR through varying a range of classification thresholds.

Please give more details of how the binary landslide and non-landslide classification related to four classes of landslide susceptibility indexes: high, moderate, low, and very low susceptibility. Is there a way to evaluate the four classes' predictions?

Some future works might be added to this manuscript. For example, integrating additional data sources, or exploring other geographical settings.

Are there any insights that can help researchers in this field? For instance, can this study be served as a foundation for others to build upon? Or can this study inspire further investigation and innovation in landslide susceptibility assessment?

Author Response

(The authors gave the same response as above.)

Round 2

Reviewer 1 Report

All comments are provided in a PDF file.

Minor editing of English language required.

Round 3

Reviewer 1 Report

Scientific work has been significantly improved both in terms of the clarity of the methods and in terms of the validity of the presentation of the results. Accept in present form.

Minor editing of English language required.